# Adequacy Analysis of the Basic Old-Age Pension System Based on Local Administrative Data in China

**Qing Zhao [1]**, **Zhen Li [2],*** **and Yihuan Wang [2]**

1   Center for Social Security Studies, Wuhan University, Hubei 430072, China; zhao.qing@whu.edu.cn
2   School of Public Administration and Policy, Renmin University of China, Beijing 100872, China; wangyihuan@ruc.edu.cn
*   Correspondence: lizhen@ruc.edu.cn

**Abstract:** There is no consensus on the judgment of the adequacy status of the old-age pension benefit in China at present. Therefore, clarification of various types of indicators and benchmarks of pension adequacy is urgently needed. According to the theoretical development of pension adequacy, this paper offers a comprehensive analysis of the benefit level of basic pension from the perspectives of poverty alleviation, income substitution, and financial sustainability. The calculation results based on local administrative data show that the current pension benefit in urban China is unbalanced: on the one hand, the average pension level of self and flexible employees cannot keep track of the local average consumption level or even the relative poverty standard in particular years and the individual replacement rates for a few nonstandard employees are less than the minimum standard of 40% set by the International Labor Organization, which means the pension benefit performs poorly in terms of consumption smoothing. On the other hand, the lifelong pension rights are much higher than the lifelong contribution obligations for new retirees. Under the trend of population ageing, the extremely high benefit–cost ratio means that the current retired generation is eroding the welfare of the current working generations, and the long-term financial sustainability of the pension system is facing challenges. In the future, in order to improve the benefit level of the basic old-age pension system in a sustainable way, we need to increase the average and individual replacement rates and reduce the benefit–cost ratio by consolidating contribution bases and delaying the number of contribution years.

**Keywords:** basic old-age pension system in urban China; average replacement rate; individual pension replacement rate; benefit–cost ratio; local administrative data

## 1. Introduction

Against the background of an ageing population and declining fertility rates, most studies have focused on pension sustainability [1–4], while relatively few adequacy analyses of the pension system have been conducted. Pension adequacy is closely correlated with people's well-being and the social sustainability of a pension system [5,6]. Thus, it is necessary to research the adequacy of pension benefits on the basis of the actual situation of the pension system in a particular country.

In China, concerns about pension benefits have never faded away. Though there are officially defined "basic benefits" and "moderate benefits" at the policy level, there is a controversy on the degree to which a pension benefit is appropriate. There are scholars who argue that the benefit of the basic old-age pension system continues to decline, because the average replacement rate, namely the average pension benefit as a percentage of the average wage, keeps decreasing by more than 70% at the initial construction of the pension system to less than 40% at present [7]. However, based on the fact that the average contribution base is only 60%–70% of the average wage, the Ministry of Human Resources and

Social Security holds that the average pension benefit as a percentage of the average contribution base could actually reach 67%, which is not low from the perspective of fairness between pension rights and obligations [8]. Many researchers have adopted the widely used "individual replacement rate" as a standard to evaluate the average replacement rate in China and have mistakenly concluded that the current benefit level is relatively low [9–11]. By simulating the individual replacement rate, others have found that the pension benefit is no less than the international standard [12]. It is clear that no consensus has been achieved so far on the status of current pension benefits. Therefore, the analysis of fundamental issues, such as "what are the indicators used to evaluate pension adequacy" and "what criteria should be used for references", is urgently needed. Otherwise, any misjudgment of the pension benefit level will lead to a sustainability crisis or adequacy predicament under inappropriate policy interventions in the future.

The remainder of the paper is structured as follows: In Section 2, we review the related literature on pension adequacy and its indicators. Section 3 provides a description of local administrative data and the practice of the pension system of one city in Central China. Section 4 calculates the average pension replacement rate, which includes two indicators. Section 5 calculates the individual pension replacement rate. Section 6 calculates the benefit–cost ratio from a lifecycle perspective. The last part displays the main findings and the conclusion.

## 2. Literature Review on Pension Adequacy

Empirical studies on the "level of pension benefits" should be traced back to the response to the primary objectives of the pension system. The World Bank (2005) outlined the primary goals of mandatory public pension schemes as adequacy, affordability, sustainability, and robustness [13]. Among these, "adequacy" refers to the retirement income provided by the pension system to prevent old-age poverty in terms of the absolute level, as well as the smoothed lifetime income in terms of the relative level. In addition, the goal of adequacy must also ensure that pension systems provide protection against longevity risk for those who live longer than the average person. The European Commission (2006) further emphasizes the relationship between "adequacy" and "sustainability" in the EU's pension assessment report, noting that the correlation between contributions and benefits and the management of longevity risks should be viewed from a lifecycle perspective, which has important implications for ensuring adequate replacement of the pension income [14]. The European Commission (2018) suggests that the "pension adequacy triangle" should include three dimensions, income maintenance, poverty prevention, and pension duration, especially in the context of an ageing population, and special attention should be paid to the balance between adequacy and sustainability within pension systems [15].

Indicators of the adequacy of pension benefits have developed from the replacement rate to the pension wealth. The replacement rate is the most frequently used indicator to reflect pension adequacy. There are various classifications based on different criteria. According to data sources, there are theoretical, simulated, and empirical replacement rates. According to time dimensions, there are horizontal and vertical replacement rates. According to whether summation occurs or not, there are individual and average replacement rates. According to whether pension income is taxed, there are gross and net replacement rates. Among these factors, the average pension replacement rate or the horizontal replacement rate are frequently used for international comparisons, in particular, by comparing the average replacement rate with poverty lines to measure the effect of the pension on reducing the poverty risk of the elderly. There is an agreement on the indicators that are used to monitor the goals of the Open Method of Coordination (OMC) in the EU, including the most importantly adequacy and sustainability of pension systems. "The average pension benefit as a percentage of the average wage" is also used by the European Commission for conducting international comparisons. The average public pension benefit is calculated by dividing the total pension expenditure by the total number of pensioners, while the average wage is calculated by dividing the total output by the total number of contributors [16]. In China, the average replacement rate is represented by the mean

pension benefit and the mean wage for some researchers [7,17–19]; meanwhile, it is represented by the median pension benefit and the median wage because of the skewed distribution of income for others [12,20,21].

In the international comparison, the individual pension replacement rate of a typical worker is also widely applied [22–24], which is both a longitudinal and theoretical indicator. To assess the individual replacement rate and make it internationally comparable, the EU sets a typical worker an adult man with an average wage pattern who retires at age 65 after 40 years of continuous contribution, as the base scenario [25]. The Organization for Economic Co-Operation and Development (OECD) also calculates the individual replacement rate of an assumed typical worker and simulates various scenarios, like 0.5 and 1.5 times that of the average wage workers. However, the theoretical replacement rate of the assumed worker is questioned by individual heterogeneity and sample representativeness. Oversimplified theoretical assumptions on individual wage profiles, contribution years, retirement age, and pension indexation rules are often inconsistent with the complicated reality, e.g., women's part-time employment, resulting in an overestimation of the actual replacement rate [26]. According to the social security agency in the US, there is an obvious gap between the individual replacement rate based on a typical worker and the actual replacement rate based on Health and Retirement Study data. There will be a 15% overestimation of the theoretical replacement rate, since the actual wage patterns are usually lower than the average wage pattern under the base scenario [27]. As a result, the theoretical replacement rate is developing in two directions. First, theoretical assumptions will be enriched to be closer to the actual situation of the population and economy. Second, administrative data from governments or income survey data from households could be used for measurements. The difficulty faced by the first direction is that the assumptions of individuals in different situations only reflect one deviation from the base case, which can neither exhaust all cases nor obtain the comprehensive situation of the replacement rate by the calculation of different weights. In the meantime, the difficulty in the second direction lies in the availability and quality of the data.

Both the average and individual replacement rates are measured based on a particular point in time. If pension adequacy is limited to a single point in time, the impact of changes, such as longevity risk, intergenerational relationships, and the institutional and economic environment, might be ignored. When comparing the levels of pension benefits among countries, the financial burden of a country with a lower life expectancy but a higher pension replacement rate is virtually the same as that of a country with a higher life expectancy but a lower pension replacement rate [28]. Therefore, it is limited to the reflection of pension adequacy only in the case of the replacement rate at a single point in time. To solve this problem, Grech (2013) proposed that it is more appropriate to estimate pension wealth, namely, the discounted present value of future pension rights, because it takes into account the payment period of future pension and measures the cash flow of the whole period, rather than at a single point [29].

In order to better reflect the adequacy of the pension to prevent poverty, smooth income, and provide financial sustainability, the internationally used indicators of pension benefits have evolved from a single-point-in-time replacement rate to an indicator based on the "working-receiving period". In China, however, previous studies have mainly focused on the single-point replacement rates, paying little attention to the "contribution-payment" phase and lacking a systematic comparison of various indicators when analyzing pension adequacy. Moreover, due to the inaccessibility of data, theoretical replacement rates are mostly simulated, while individual replacement rates based on actual data are presented less often. In view of this, by using local administrative data composed of annual contributions and reception records of the insured retirees, this research calculates different kinds of adequacy indicators and refers them to the corresponding criteria in order to evaluate the pension adequacy status from multidimensional perspectives, thus providing policy implications for the adequate and sustainable development of the basic old-age pension system in the future.

## 3. Basic Old-Age Pension System of City X in Central China

The "basic old-age pension system" in this paper only refers to the employment-related public pension system with a history of more than twenty years. It is a typical Bismarck model which could be used for international comparison. The residents' public pension system is heavily subsidized by governments, and the pension benefit merely constitutes a tiny share of pensioners' income, and therefore it is not discussed in this paper. Due to the independence of the funding pool, the civil servants' retirement pension is also outside the discussion. In this research, city X of Central China is selected as a sample for the following reasons: Firstly, data is only available from the social insurance agency of city X because of a collaboration project between the government and our academic institutes. Secondly, city X is a representative sample of a large number of central and western cities in China. As for economic development, in 2017, city X's per capita GDP was 29,308 yuan, lower than the national average level of 59,201 yuan. As for the population, the total population of city X is 9.74 million, among which 4.21 million are urban residents. The urbanization rate of the resident population is about 48%, which is lower than the national average of about 60%. In terms of people's living standards, the Engel coefficient of urban residents' households is about 25%—lower than the national average level.

### 3.1. Institutional Characteristics of the Basic Old-Age Pension System in City X

Under the guidance of document No. 26 issued by the State Council in 1997, City X established basic old-age social insurance for urban employees to replace the traditional urban retirement system. The newly constructed pension system, which is a combination of the Pay-As-You-Go social pooling part and the personal account, originally covered employees in state-owned enterprises [30]. Since 2005, guided by the No. 38 document issued by the State Council, the local government of city X has expanded the coverage of the urban public pension system to nonpublic enterprises, self-employees, and flexible employees and has continued to include farmers who lost land during the construction of industrial estates from 2006. To encourage more participants from informal sectors, the policy not only sets lower contribution rates and provides contribution bases that can be chosen, but also allows for lump-sum contributions and delayed contributions, even when people reach the statutory retirement age. In this way, there are three categories of participants within the pension system: enterprise employees, nonstandard employees, and land-lost farmers. The latter two are participants as individuals (without employers), who use lower contribution rates but the same benefit formula as enterprise employees.

### 3.2. Data and Descriptive Statistics

The administrative data provided by the local social insurance agency contain all participants' historical contribution and benefit records. By the end of 2018, there were nearly 19 thousand pensioners with more than three million records, among which 39.45% were males and 60.55% were females. In terms of the employment type, enterprise employees accounted for only 14%, nonstandard employees accounted for 57%, and land-lost farmers accounted for the remaining 29%. Regarding different benefit formulas, the "old" retirees, referring to those who retired before the establishment of the old-age social insurance in 1997, constitute 0.27% of the total retirees; the "old medium", which refers to those who worked before 1997 but retired between 1997 and 2005, constitutes 1.7%; the "new medium", referring to those who began working before 1997 but retired after 2005, constitutes 40.88%; and the "new" retirees, meaning those who began working after 1997, constitute 57.15% of the retired population.

With respect to the historical contribution records, the average number of accumulated contribution years is 21.73 and 26.32 years for males, while it is 18.75 years for females. Moreover, enterprise employees have the greatest number of accumulated contribution years, 32.02 on average, followed by nonstandard employees with 22.52 years on average, and land-lost farmers with the least—15.13 years on average. As can be seen from Figure 1, most of the individual participants (including nonstandard employees and land-lost farmers) stop contributing after reaching 15 years, the regulated minimum

contribution period. It is clear that noncontinuous contribution is quite prevalent in this city, and this violates the general assumption that "people keep contributing from the day they start work".

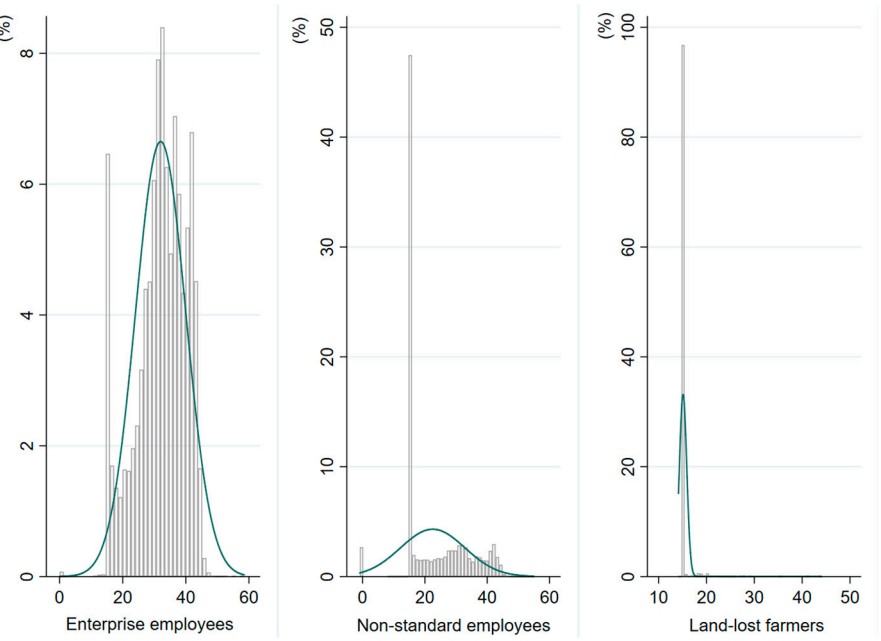

**Figure 1.** The distribution of accumulated contribution years by different types of retirees.

According to Figure 2, the average contribution base of basic old-age insurance for retirees has gradually increased from 285 yuan per month in 1998 to 3167 yuan per month in 2018, with an average annual growth rate of about 12.54%. The contribution base for enterprise employees is usually higher than that of nonstandard employees and land-lost farmers. According to Figure 3, the average monthly pension benefit has increased from 337 yuan in 1998 to 1923 yuan in 2018, with an average pension growth rate of 9.32%, which is lower than the growth rate of the average contribution base during the same period.

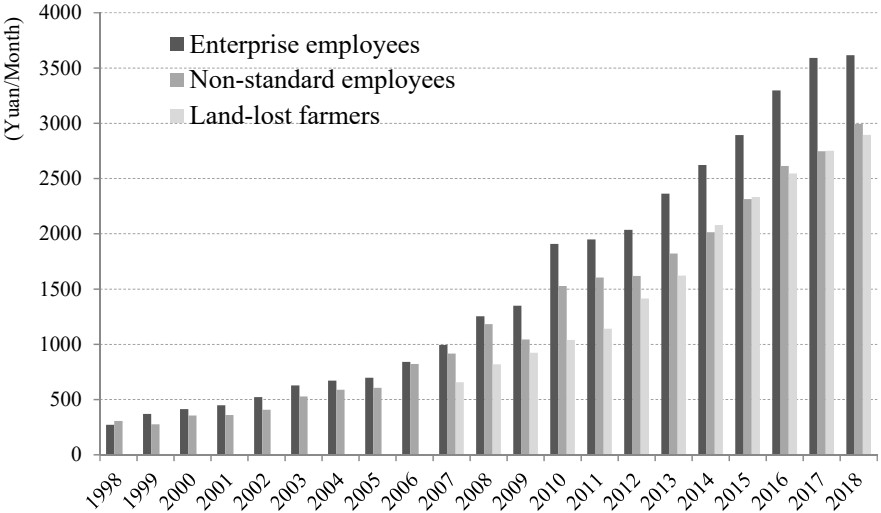

**Figure 2.** Average contribution bases by various types of retirees in the last two decades.

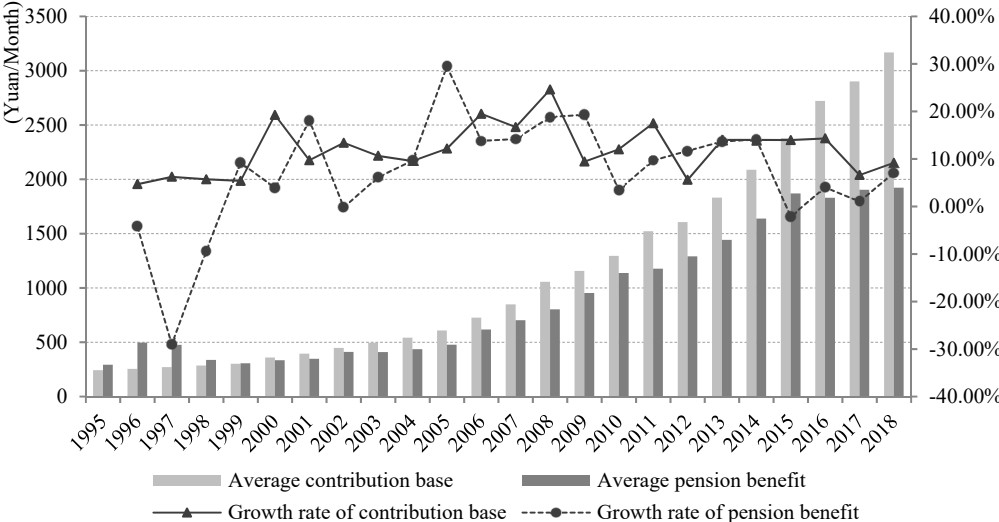

**Figure 3.** Changes in the average contribution bases, average pension benefits, and their growth rates the last two decades.

## 4. The Average Replacement Rate Based on a Particular Point of Time

The average replacement rate, also known as the horizontal replacement rate, depicts the relationship between the retirement income and working income in the society at a particular point in time. In the absence of micro-survey data, the average pension replacement rate based on macro data is often used to evaluate the average level of pension benefits. In the following section, we calculate this indicator from two aspects: the average wage and the average contribution wage/base.

### 4.1. Average-Wage-Replacement Rate

The average-wage-replacement rate is used to reflect the relative situation between the pension benefits of the retired population and the wage income of the working population. It is also an indicator to describe the relative economic situation of the elderly in a society or the degree of intergenerational solidarity [31]. The average-wage-replacement rate of the basic old-age pension in China is usually expressed as follows:

$$RR1 = \frac{\overline{B}_t}{\overline{W}_{t-1}} \tag{1}$$

where $\overline{B}_t$ stands for the average pension benefit of all pensioners in year $t$, and $\overline{W}_{t-1}$ stands for the average wage of the previous year.

The average-wage-replacement rates of city X over the years based on the local social security bulletin and administrative data are shown in Table 1. From 1999 to 2018, the average replacement rate had a downward trend on the whole, from more than 70% at the beginning of 2000 to less than 40% in 2018, although there was a rebound between 2006 and 2010. The declining trend of this indicator could be largely explained by the fact that the average growth rate of nominal wages from 1999 to 2018 on average is around 14.5%, which is much higher than the pension indexation rate in corresponding years. It is worth noting that the average wage in China now is based on the average salary of urban employees from formal sectors. If the statistical caliber of the average wage is changed to the wages of employees from both formal and informal sectors, this indicator will rise.

In China, researchers usually judge the reasonable level of pension benefits based on the basic living needs of the elderly after retirement. Zheng (2003) argued that the average replacement rate of 50% based on a resident consumption level with an Engel coefficient of 0.4 is adequate and predicted that the replacement rate could be lower with the decline of the Engel coefficient [32]. Li and Wang (2012) stated that the Engel coefficient only ensures residents' food needs, whereas basic living needs

for the elderly should also include clothing, housing, transportation, communication, and health care [18]. Bian and Sun (2018) extended the basic needs of the elderly from material to spiritual aspects, including cultural and educational expenditure [33]. This study compared the average pension benefit with the per capita consumption expenditure of urban residents in the previous year, finding that the ratio is between 1 and 1.5, which means that the average pension benefit can satisfy the basic material and spiritual needs of the elderly.

Internationally speaking, the relative poverty line is generally set as 50% of the median income. Because of lack of wage-income distribution data, the mean disposable income of urban residents is used to replace median income here. As is shown in Table 1, the ratio of average pension benefits to the relative poverty line has fluctuated over the past two decades, and it is always 1.5 times more than the relative poverty line. However, this ratio can be underestimated if the median income is used, because the mean income is always larger than median income in the thick-tailed distribution. Overall, although the average-wage-replacement rate in city X presents a downward trend, the average pension benefit is higher than the local average consumption level and the relative poverty line.

Specifically speaking, the pension benefits of enterprise-employees are generally better than those of nonstandard employees and land-lost farmers. In particular years, the pension benefits of individual participants did not catch up with the average consumption expenditure, and land-lost farmers fell into a relative poverty trap, as is demonstrated in Table 2.

### 4.2. Average-Contribution Base-Replacement Rate

In China, policies on contribution bases are not consistent, and there are huge differences in the determinants of contribution bases from province to province. In practice, there is one standard for employees and another for employers [34]. In addition, there are usually multiple grades of contribution bases independent of income for individual participants to choose from. Since there is generally a gap between the contribution base and the actual wage income in China, it is necessary to supplement the indicator of the average-contribution base-replacement rate to reflect the relative relationship between the pension entitlement and the actual contribution at a particular point in time. The indicator is shown below:

$$RR2 = \frac{\overline{B}_t}{\overline{CW}_{t-1}} \qquad (2)$$

where $\overline{CW}_{t-1}$ stands for the average contribution wage/base in the previous year.

According to the administrative data, the average contribution base in city X in previous years was generally lower than the average wage. The proportion of the contribution base to the average wage gradually decreased from over 70% in 1999 to 57% in 2018 (see Table 3). Due to the shrinking of the contribution base, the average-contribution base-replacement rate is much larger than the average-wage-replacement rate. It can be seen that the average pension benefit is relatively high compared to the average contribution base in this city. Only when the contribution wage is in line with the average wage will this indicator and the average-wage-replacement rate be equal.

**Table 1.** Comparison of the average pension benefits with the average wage, average consumption level, and relative poverty line.

| Year | Average Pension Benefits/Average Wages in the Previous Year | Average Pension Benefits/Average Consumption Expenditure of Urban Residents in the Previous Year | Average Pension Benefits/50% per Capita Disposable Income in the Previous Year | Year | Average Pension Benefits/Average Wages in the Previous Year | Average Pension Benefits/Average Consumption Expenditure of Urban Residents in the Previous Year | Average Pension Benefits/50% per Capita Disposable Income in the Previous Year |
|---|---|---|---|---|---|---|---|
| 1999 | 78.28% | 1.14 | 1.87 | 2009 | 62.08% | 1.29 | 1.93 |
| 2000 | 70.79% | 1.10 | 1.74 | 2010 | 65.81% | 1.38 | 2.12 |
| 2001 | 63.06% | 1.20 | 1.73 | 2011 | 59.81% | 1.32 | 1.99 |
| 2002 | 65.02% | 1.22 | 1.86 | 2012 | 55.76% | 1.33 | 1.93 |
| 2003 | 57.46% | 1.03 | 1.59 | 2013 | 52.62% | 1.36 | 1.85 |
| 2004 | 56.03% | 1.02 | 1.55 | 2014 | 48.04% | 1.47 | 1.89 |
| 2005 | 55.42% | 1.04 | 1.55 | 2015 | 49.69% | 1.53 | 1.96 |
| 2006 | 62.63% | 1.24 | 1.81 | 2016 | 43.63% | 1.37 | 1.76 |
| 2007 | 64.24% | 1.22 | 1.84 | 2017 | 41.33% | 1.35 | 1.69 |
| 2008 | 59.31% | 1.19 | 1.83 | 2018 | 38.22% | 1.24 | 1.56 |

**Table 2.** Comparison of the average pension benefits with the average wage, average consumption level, and relative poverty line by different types of retirees.

| Year | Average Pension Benefits/Average Wages in the Previous Year | | | Average Pension Benefits/Average Consumption Expenditure of Urban Residents in the Previous Year | | | Average Pension Benefits/50% per Capita Disposable Income in the Previous Year | | |
|---|---|---|---|---|---|---|---|---|---|
| | Enterprise Employees | Nonstandard Employees | Land-Lost Farmers | Enterprise Employees | Nonstandard Employees | Land-Lost Farmers | Enterprise Employees | Nonstandard Employees | Land-Lost Farmers |
| 2008 | 57.79% | 51.57% | 26.92% | 1.16 | 1.03 | 0.54 | 1.78 | 1.59 | 0.83 |
| 2009 | 56.91% | 53.30% | 25.48% | 1.19 | 1.11 | 0.53 | 1.77 | 1.66 | 0.79 |
| 2010 | 67.06% | 63.91% | 25.07% | 1.41 | 1.34 | 0.53 | 2.16 | 2.05 | 0.81 |
| 2011 | 59.68% | 44.78% | 25.58% | 1.32 | 0.99 | 0.57 | 1.98 | 1.49 | 0.85 |
| 2012 | 57.08% | 34.95% | 46.25% | 1.37 | 0.84 | 1.11 | 1.97 | 1.21 | 1.60 |
| 2013 | 56.76% | 50.57% | 25.86% | 1.47 | 1.31 | 0.67 | 1.99 | 1.78 | 0.91 |
| 2014 | 51.69% | 33.89% | 19.45% | 1.58 | 1.04 | 0.60 | 2.03 | 1.33 | 0.77 |
| 2015 | 49.04% | 30.62% | 40.37% | 1.51 | 0.94 | 1.24 | 1.93 | 1.20 | 1.59 |
| 2016 | 47.81% | 28.01% | 18.86% | 1.50 | 0.88 | 0.59 | 1.92 | 1.13 | 0.76 |
| 2017 | 50.77% | 27.26% | 23.43% | 1.66 | 0.89 | 0.77 | 2.07 | 1.11 | 0.96 |
| 2018 | 47.73% | 27.49% | 18.77% | 1.55 | 0.90 | 0.61 | 1.95 | 1.12 | 0.77 |

**Table 3.** The average-contribution base-replacement rate over the years.

| Year | Average Contribution Base/Average Wage | Average Pension Benefit/Average Contribution Base in the Previous Year | Year | Average Contribution Base/Average Wage | Average Pension Benefit/Average Contribution Base in the Previous Year |
|---|---|---|---|---|---|
| 1999 | 73.12% | 107.05% | 2009 | 68.78% | 90.25% |
| 2000 | 63.80% | 110.95% | 2010 | 66.88% | 98.39% |
| 2001 | 65.22% | 96.68% | 2011 | 65.81% | 90.89% |
| 2002 | 62.47% | 104.07% | 2012 | 65.71% | 84.85% |
| 2003 | 62.71% | 91.63% | 2013 | 58.64% | 89.73% |
| 2004 | 63.72% | 87.93% | 2014 | 53.70% | 89.45% |
| 2005 | 62.98% | 87.99% | 2015 | 55.52% | 89.49% |
| 2006 | 61.66% | 101.58% | 2016 | 56.76% | 76.88% |
| 2007 | 66.41% | 96.74% | 2017 | 59.09% | 69.94% |
| 2008 | 62.66% | 94.65% | 2018 | 57.65% | 66.30% |

Specifically speaking, the proportion of the average contribution base to the average wage for enterprise employees is usually higher than that of the other two types, and so is the average-contribution base-replacement rate, as shown in Table 4.

**Table 4.** The average-contribution base-replacement rate by different types of retirees.

| Year | Average Contribution Base/Average Wage | | | Average Pension Benefit/Average Contribution Base in the Previous Year | | |
|---|---|---|---|---|---|---|
| | Enterprise Employees | Nonstandard Employees | Land-Lost Farmers | Enterprise Employees | Nonstandard Employees | Land-Lost Farmers |
| 2008 | 73.52% | 67.71% | 48.58% | 78.61% | 76.16% | 55.40% |
| 2009 | 81.63% | 76.99% | 53.31% | 69.72% | 69.23% | 47.80% |
| 2010 | 78.10% | 60.34% | 53.44% | 85.86% | 65.91% | 46.92% |
| 2011 | 97.00% | 77.60% | 52.81% | 61.53% | 57.71% | 48.43% |
| 2012 | 84.15% | 69.27% | 49.27% | 67.83% | 50.46% | 93.88% |
| 2013 | 74.30% | 59.06% | 51.60% | 76.39% | 85.62% | 50.11% |
| 2014 | 69.30% | 53.38% | 47.52% | 74.59% | 63.49% | 40.93% |
| 2015 | 69.73% | 53.54% | 55.27% | 70.33% | 57.19% | 73.04% |
| 2016 | 69.01% | 55.18% | 55.63% | 69.27% | 50.76% | 33.91% |
| 2017 | 71.61% | 56.75% | 55.28% | 70.90% | 48.03% | 42.39% |
| 2018 | 71.38% | 54.58% | 54.66% | 66.86% | 50.37% | 34.33% |

## 5. The Individual Replacement Rate at the Point of Retirement

The individual replacement rate, also known as the longitudinal replacement rate, reflects the preretirement income and postretirement income of a particular individual. This indicator is commonly used to measure the degree to which pension benefits can maintain the living standard of an individual or the degree to which the consumption is smoothed. In general, 40% (ILO, 1952 [35]), 45% (ILO, 1967 [36]), and 55% (ILO, 1967 [37]) are used as the reference standards, as defined by the three conventions of the International Labor Organization. The targeted replacement rates of the basic old-age pension system of 58.5% [38] set by document No. 26 in 1997 and 59.2% [39] set by document No. 38 in 2005 are both theoretical replacement rates for typical workers. Because of the deviation between theory and reality, the targeted replacement rate can be overestimated [40]. In an attempt to develop the indicator from the theoretical replacement rate to the empirical replacement rate, we calculated the individual replacement rate based on local administrative data, so as to reflect the heterogeneous characteristics of the actual pension benefits.

### 5.1. Formulas for the Individual Replacement Rate

We used contribution bases to substitute real wages because of missing data. The individual replacement rate is displayed as follows:

$$RR3 = \frac{B_r^1 + B_r^2 + B_r^3}{CW_{r-1}} \tag{3}$$

where $B_r^1$ represents the basic pension for the first year of retirement; $B_r^2$ represents the personal account pension for the first year of retirement; $B_r^3$ represents the transitional pension for the first year of retirement; and $CW_{r-1}$ represents the personal contribution wage for the year prior to retirement. When the retirees are "new" members, who join after the new pension system is established, their transitional pension is 0.

Moreover, $e$ was assumed to be the age at joining the pension scheme, $r$ is the retirement age, m is the accumulated contribution years (including the years perceived as "contributing", used as $m_0$), $n$ is the preset benefit months, $\alpha$ is the rate of contribution to the personal account, $\gamma_j$ is the interest rate of the personal account, $CW_i$ is the individual contribution base in year i, and $\overline{S}_{t-1}$ stands for the average wage a year before an individual's retirement. According to the pension reform policy of document No.38 issued in 2005, the benefit formulas for the "new" and "new medium" retirees are in as follows:

$$B_r^1 = \frac{1\% \cdot m \cdot \overline{S}_{t-1}}{2} \left(1 + \frac{1}{r-e-1} \cdot \sum_{i=t-r+e}^{t-1} \frac{CW_{i,i+r-t}}{\overline{S}_{i-1}}\right) \tag{4}$$

$$B_r^2 = \frac{\alpha}{n} \cdot \sum_{i=t-r+e}^{t-1} \left(CW_{i,i+r-t} \cdot \prod_{j=i}^{t-1}\left(1 + \gamma_j\right)\right) \tag{5}$$

$$B_r^3 = \frac{\overline{S}_{t-1}}{r-e-1} \cdot \left(\sum_{i=t-r+e}^{t-1} \frac{CW_{i,i+r-t}}{\overline{S}_{i-1}}\right) \cdot m_0 \cdot 1.1\% \tag{6}$$

For the "old medium" retirees, the benefit formulas set by the pension reform policy of document No. 26, issued in 1997, are as follows:

$$B'^1_r = 20\% \cdot \overline{S}_{t-1} \tag{7}$$

$$B'^2_r = \frac{\alpha}{120} \cdot \sum_{i=t-r+e}^{t-1} \left(CW_{i,i+r-t} \cdot \prod_{j=i}^{t-1}\left(1 + \gamma_j\right)\right) \tag{8}$$

$$B'^3_r = \frac{\overline{S}_{t-1}}{r-e-1} \cdot \left(\sum_{i=t-r+e}^{t-1} \frac{CW_{i,i+r-t}}{\overline{S}_{i-1}}\right) \cdot m_0 \cdot 1.2\% \tag{9}$$

### 5.2. Results Analysis

The values of $B_r^1$, $B_r^2$, $B_r^3$, and $CW_{r-1}$ are directly provided by the administrative data of records in city X; therefore, the individual replacement rate of the first-year pension benefit to the contribution wage prior to retirement can be computed. The individual replacement rate is, on average, 49%–63.93% for men and 39.27% for women. The distribution of the indicator by gender is shown in Figure 4. From the perspective of retiree type, the individual replacement rate for enterprise employees is 73.53%, on average, among which the rate of males is 86.91% and that of females is 57.81%. This is followed by nonstandard employees at 51.47% and land-lost farmers at 32.04%, on average (see Figure 5). It can be seen that the individual replacement rate of enterprise employees is generally higher than that of individual participants, and men generally have higher rates than women. According to the minimum standard of the individual replacement rate set by the ILO convention, which states that "a typical worker's pension replacement rate should be no less than 40%", the female nonstandard employees

(around 40%), as well as land-lost farmers (32%), are perceived to be inadequate pensioners because of their poor pension functions in consumption smoothing. However, it is also worth noting that the value of the individual replacement rate would be lower if the personal wage was used instead of the contribution wage.

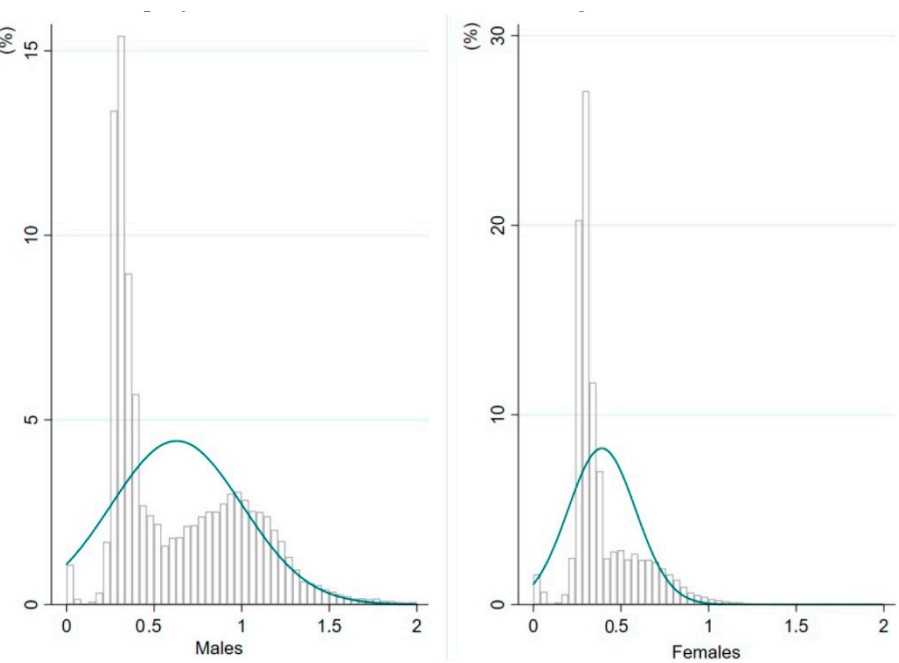

**Figure 4.** Distribution of the individual replacement rate by gender.

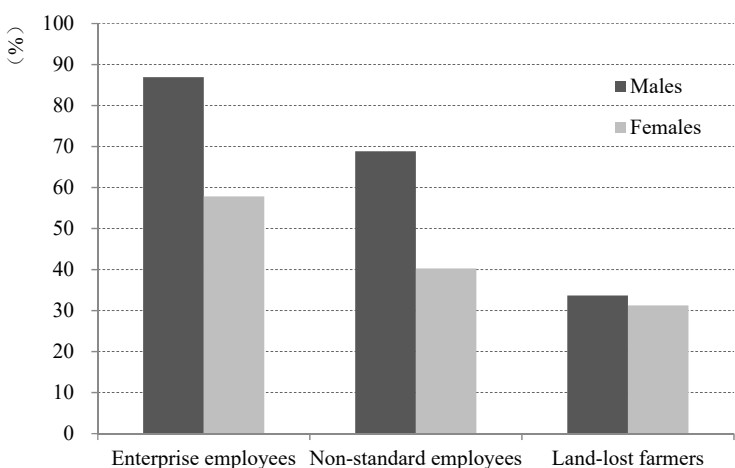

**Figure 5.** Comparison of the individual replacement rate by different types of retiree.

The disparities of individual replacement rates among different groups can be explained by benefit formulas. Since 97% of the retired population belongs to the "new" and "new medium" groups, their pension benefit formula follows the 2005 pension-reform policy. According to the formula, the accumulated contribution years and contribution wages are the main determinants of the basic pension gap, and the contribution wages also determine the personal-account pension and transitional pension among individuals. Other things being equal, the more contribution years and the bigger the contribution base, the higher the pension benefits are. As is demonstrated in Figure 1, the number of accumulated contribution years of enterprise employees is generally greater than that of individual participants, and the annual contribution bases for enterprise employees are generally larger than those of individual participants (see Figure 2); also, men usually have more contribution years and larger

contribution bases than their female counterparts. Consequently, the individual replacement rate for enterprises employees is generally higher than that of the rest, especially for female nonstandard employees and land-lost farmers (see Figure 5).

## 6. The Benefit–Cost Ratio Based on Individual Lifecycle

Changes in the longevity risk and indexation of pension benefits will not be reflected in the replacement rate at a point of time, but rather, will be reflected in the pension wealth. Given this situation, we adopted the benefit–cost ratio indicator, namely the ratio of the present value of individual lifetime pension benefits to the final lifetime contribution value to describe the relevant relationship between benefit rights and contribution obligations within an individual's lifecycle.

### 6.1. Basic Assumptions

(1)　Pension growth rates: Based on the growth rate of the average pension benefit over the years and the forecast for future economic growth [41,42], it is assumed that the average annual growth rate of pensions in the future will be 5%. In this case, pension benefits for each individual after 2018 can be computed by using the pension benefits in previous years.

(2)　The interest rates: The interest rate for personal accounts in the basic old-age pension system is the one-year bank-deposit rate, and it is assumed to remain at the current level of 4% [43].

(3)　Life expectancy: An age- and gender-specific life table was constructed in accordance with Zeng's (2016) simulation of urban residents in China [44], and the age limit for both men and women was assumed to be 101 years old.

### 6.2. Actuarial Models

Given the basic assumptions, the individual benefit–cost ratio of the pension system in city X can be expressed as follows:

$$RR4 = \frac{PV}{FV} \tag{10}$$

$$FV = c \cdot \sum_{j=e+1}^{r-1} CW_j \cdot (1+i)^{r-j} \tag{11}$$

$$PV = \sum_{k=r+1}^{\omega-1} B_k \cdot (1+v)^{\omega-k} \tag{12}$$

$$B_k = \begin{cases} B_t & t \le k,\ k \le 2018 \\ B_{2018} \cdot (1+f)^{\omega-k-1} & k > 2018 \end{cases} \tag{13}$$

where *PV* represents the present value of lifetime pension benefits, and *FV* represents the final value of lifetime contributions for each individual; *r* stands for the retirement age; *e* stands for the age at joining the pension scheme; $\omega$ stands for the age limit; *c* stands for the contribution rate set by policies, i.e., 28% for enterprise-employees and 20% for nonstandard employees and land-lost farmers; *i* stands for the risk-free interest rate; *v* is the corresponding discounted rate ($v = 1/(1+i)$); *f* stands for the pension growth rate; $CW_j$ stands for the contribution wage in year *j*; and $B_k$ stands for the pension benefit in year *k*.

### 6.3. Results Analysis

The benefit–cost ratio cannot be calculated for "old" retirees, because they do not have contribution records under the new pension system. As for the "medium" retired population, there are no actual contributing records for the perceived periods; hence, the benefit–cost ratio would be overestimated if only actual contribution periods were taken into consideration. Therefore, only the analysis of

the "new" retirees with complete contribution records under the old-age social-insurance system has practical significance.

The simulated results show that the mean level of the benefit–cost ratio of "new" retirees is 7.05 and the median is 6, among which the ratio of males is 6.09 and that of females is 7.48, on average, as shown in Figure 6. In terms of retiree types, 96% of the retirees are individual participants, with nonstandard employees and land-lost farmers each accounting for about 48%. This is due to the coverage expansion policy that has existed since 2005, which encourages self-employees, flexible employees, and even land-lost farmers to participate in the urban pension system, even through lump-sum contributions or by delaying their contributions if their contribution period is less than 15 years. The individual participants constitute the majority of the newly retired population because most enterprise employees have not reached the statutory retirement age. The benefit–cost ratio by different types of retiree is displayed in Figure 7, showing average values of 7.05 for nonstandard employees and 6.74 for enterprise employees.

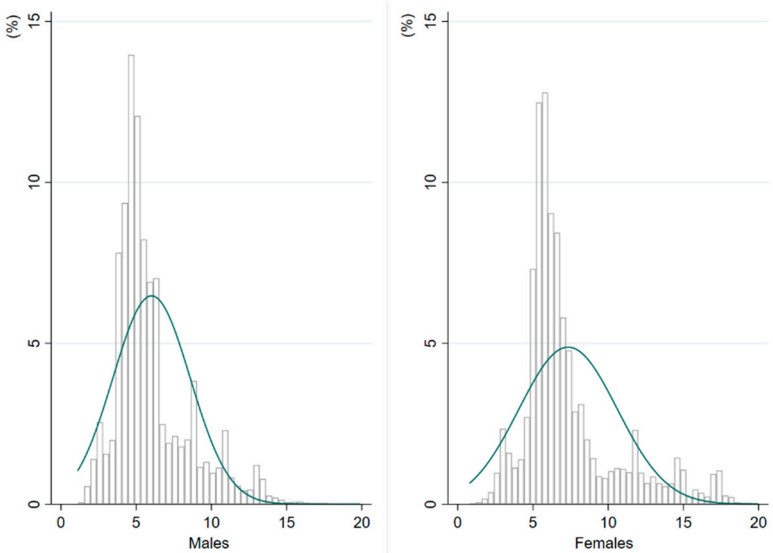

**Figure 6.** Distribution of the benefit–ratio for the "new" retired population by gender.

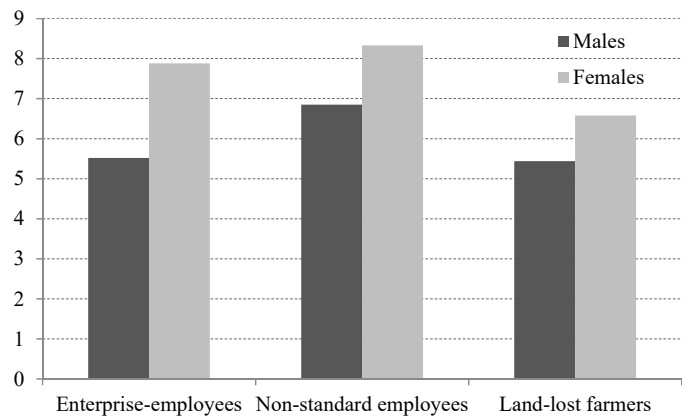

**Figure 7.** Comparison of the benefit–ratio for the "new" retired population by retirement types.

The standard value of this indicator under a balanced status between pension rights and contribution obligations is 1. When the ratio is greater than 1, it means that an individual's pension rights exceed his/her obligations to a large degree. The general situation of the pension system in city X suggests that the present value of lifetime pension benefits is seven times more than the final value of all contributions, meaning that pension obligations and pension rights are severely imbalanced.

This phenomenon could be explained by the RR4 formula, because basic pensions and transitional pensions are paid in a lifelong manner by the social pooling account within the pension system, and the personal account is still paid by the social pooling part after it is used up. This is to say, the length of pension entitlement solely depends on how long one lives after retirement. In fact, the average remaining lifespan (nearly 22 years) of the retired population in China is generally higher than the average number of contribution years and is especially higher than the minimum number of contribution years of 15. As a result, the present value of the lifelong pension benefit is usually higher than the final value of lifelong contributions given particular pension growth rates and interest rates. With respect to the differences among different groups, the annual pension benefit does not contain the differentiated contribution rates between enterprise employees and others, so the present value of pension benefits between these groups is smaller; however, the final value of contributions does contain the differentiated contribution rates, contribution bases, and contributing periods, and hence, there are larger gaps between various employment groups. This is the reason why the benefit–cost ratio for nonstandard employees with lower contribution rates, bases, and periods is usually higher than that for enterprise employees. In addition, with other conditions being equal, the benefit–cost ratio is higher for females than males, because of their lower retirement age and longer life expectancy.

Under the circumstances of a younger age structure and sustained economic growth, the working generation is able to support the older population with a high benefit–cost ratio within the Pay-As-You-Go pension system. However, the proportion of people over 60 years old within the total population in China is projected to double around 2050, exceeding 30% [45], indicating the coming of an aging society. In this context, the working generation will not be able to the afford pension expenditure of the retired generation with such a high benefit–cost ratio. In other words, the system is facing a sustainability crisis. Thus, a core issue of future policy reform is how to balance pension rights and obligations. Using the formula of the benefit–cost ratio, we found that the interest rate and its corresponding discount rate are closely related to the unpredictable macroeconomic environment, which does not allow easy intervention. Reducing the growth rate of pension benefits can reduce the present value of pension benefits in theory, but it is often faced with the challenges of welfare rigidity and political pressure in reality. Compared with the adjustment of other parameters, the imbalance between pension rights and contribution obligations can be modified by increasing the number of contribution years and strengthening the contribution bases in the short run, as well as conducting reforms like linking pension rights to life expectancy, for example, in the long run. Nevertheless, it is worth noting that increasing the contribution period and consolidating the contribution base are also conducive to increasing the individual replacement rate and improving the consumption smoothing function for the elderly, which is particularly important for nonstandard female employees and land-lost farmers.

## 7. Conclusions

Based on the latest developments in pension adequacy, this paper evaluated the benefit level of the basic old-age pension system in urban China from the perspectives of poverty alleviation, income substitution, and financial sustainability. Different from previous calculations using macro data and simulations based on the assumptions of typical individuals, this paper used local administrative data on historical contribution and benefit records in city X of Central China to evaluate the status of pension adequacy from the single time-point and lifecycle dimensions. The main findings are as follows.

Firstly, the average pension benefit in city X, compared with the average wage, per capita consumption expenditure, and per capita disposable income, reflects the extent to which the pension benefits prevent old-age poverty and maintains the basic living needs of the elderly. Although there was a decline in the average-wage-replacement rate between 1999 and 2018, from 78% to 38%, the average pension benefit was able to keep up with the urban residents' average consumption level and was able to cross the relative poverty line. In terms of subgroups of retirees, however, the pension benefits of

nonstandard employees and land-lost farmers were lower than the local average consumption level or even the relative poverty line in particular years.

Secondly, the average contribution wage is not consistent with the average wage in China. The proportion of the contribution base in the average wage dropped from 73% in 1999 to 58% in 2018, so the "average-contribution base-replacement rate" has generally been higher than the "average-wage-replacement rate". In 2018, the average-contribution base-replacement of the pension system in city X was 66%, indicating that the average pension level was quite high relative to the average contribution base of the society. When pension rights are more closely related to contribution obligations, i.e., the contribution wage is in line with the actual wage, the two types of replacement rates tend to be equal.

Thirdly, the individual replacement rate is a longitudinal indicator that reflects the degree of consumption smoothness of pension income at the point of retirement. In this study, obvious disparities in the individual replacement rates were found between genders and various employment types. The indicator of enterprise employees was shown to be relatively high, whereas female nonstandard employees and land-lost farmers were shown to have rather low individual replacement rates, even below the 40% minimum standard set by the ILO, indicating a poorer function of consumption smoothing.

Finally, the benefit–cost ratio, which takes consideration of an individual's lifecycle factors, such as longevity risk, intergenerational relationships, and systematic changes, provides a comprehensive reflection of the relative relationship between pension rights and contribution obligations. According to historical contributing records and assumptions on future benefits, the present value of lifelong benefits is seven times more than the final value of lifetime contributions on average for the "new" retired population. It can be seen that, under the current system, pension rights and obligations are severely unbalanced for individual participants, who account for the majority of retirees. Such a high benefit–cost ratio poses a threat to the wellbeing of future generations under the trend of population aging and indicates a potential risk to the long-term sustainability of the pension system.

In conclusion, although city X is not representative of China as a whole, an analysis using local administrative data can show a common phenomenon across the less-developed regions in Central and Western China, which is significant enough to get the attention of policy makers. The pension benefit level is extremely unbalanced within the urban public-pension system. Lifetime pension rights are far more than lifetime contribution obligations for the current retired population, who have become net beneficiaries of the pension system. However, in the meantime, some nonstandard employees and land-lost farmers have to face the plight that their pension benefits at a single point of time will be too low to resist poverty risks and to realize consumption smoothing (see Table 5). Given this situation, it is of vital importance to increase the contribution period and contribution base to improve the individual replacement rate while reducing the benefit–cost ratio to realize the sustainable improvement of pension benefits.

**Table 5.** Indicators of pension adequacy for the old-age pension system of city X in 2018.

| Types of the Retired Population | Single Point of Time | | | | | Life Cycle |
|---|---|---|---|---|---|---|
| | Average Pension Benefit (with Reference to) | | | | Individual Pension Benefit at the Time of Retirement | Present Value of Lifetime Benefits |
| | The Average Consumption Level of Local Residents | The Relative Poverty Line | The Average Wage | The Average Contribution Wage | With Reference to the Contribution-Wage a Year Before Retirement | With Reference to the Final Value of Lifetime Contributions |
| All | 1.24 | 1.56 | 38.22% | 66.30% | 49.00% | 7.05 |
| Enterprise employees | 1.55 | 1.95 | 47.73% | 66.86% | 73.53% | 6.74 |
| Nonstandard employees | 0.90 | 1.12 | 27.49% | 50.37% | 51.47% | 7.89 |
| Land-lost farmers | 0.61 | 0.77 | 18.77% | 34.33% | 32.04% | 6.22 |

**Author Contributions:** Conceptualization, Z.L. and Q.Z.; data curation, Q.Z. and Y.W.; methodology, Q.Z.; validation, Z.L., Q.Z. and Y.W.; formal analysis, Z.L., Q.Z. and Y.W.; investigation, Z.L.; writing—original draft preparation, Q.Z.; writing—review and editing, Q.Z. and Y.W.; funding acquisition, Z.L.

**Funding:** This research was funded by The Key Project of National Social Science Foundation in China, grant number 15AJL012, "Research on the Construction of Equitable and Sustainable Old-Age Income Security System in China".

**Acknowledgments:** We acknowledge the support from the Deputy Director of the Social Insurance Agency in Central China for the local administrative data they provide. We also appreciate Wanding Huang, Chengyang Zhang, and Chen Chen from Renmin University of China for their technical support during the field survey.

**Conflicts of Interest:** The authors declare no conflicts of interest. The funders had no role in the design of the study; in the collection, analyses, or interpretation of data; in the writing of the manuscript, or in the decision to publish the results.

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
