# Peer review of "Adequacy Analysis of the Basic Old-Age Pension System Based on Local Administrative Data in China"

_sustainability, doi:10.3390/su11247196_

Round 1

Reviewer 1 Report

The article provides an interesting review of the adequacy. of the pension system in China, based on the administrative data collected from a city that is representative for Northern China. It provides an interesting insight on the development of pension levels, relative to wages, as well as differences in pensions between different types of pensioners as well as by gender. To that end, I assess it as providing an important contribution to the empirical knowledge on the development of pension systems in China.

To improve the quality of the article, I have several suggestions.

First, in the literature review it would be important to deepen the literature review and existing descriptions. I lacked the reference to ILO 102 Convention. Furthermore, the context of the Open Method of Coordination in the EU is not presented - this is a policy coordination mechanism, upon which the EU member states agreed on the pension goals, that include most importantly adequacy and sustainability of pensions. Based on these goals, the countries agreed on the set of monitoring indicators, including also the replacement ratios as well as individual replacement rates. So, the OMC did not introduce indicators (lines 85-86), rather there is an agreement on indicators that are used to monitor the goals of the OMC (one can look at the work of David 
Natali or Caroline de la Porte). 

In text, it would be good to expand some of the conclusions related to the presented data. For example, if the expansion of the pension system to cover farmers in 2006 explains some of the drop in average pension/average wage ratio? There is definitely a very high decline in this ratio in the past few years, which seems to be caused by the high wage growth and not following this growth the pension indexation. 

Presented figures show that the distribution of contributory years, but also pension benefits is not typical, it has one or more modes that are explained by the existing rules. Therefore, I would not fit the distribution function in figures 1,4 and 6 - it does not make an economic sense. 

Author Response

The article provides an interesting review of the adequacy. of the pension system in China, based on the administrative data collected from a city that is representative for Northern China. It provides an interesting insight on the development of pension levels, relative to wages, as well as differences in pensions between different types of pensioners as well as by gender. To that end, I assess it as providing an important contribution to the empirical knowledge on the development of pension systems in China.

To improve the quality of the article, I have several suggestions.

First, in the literature review it would be important to deepen the literature review and existing descriptions. I lacked the reference to ILO 102 Convention. Furthermore, the context of the Open Method of Coordination in the EU is not presented - this is a policy coordination mechanism, upon which the EU member states agreed on the pension goals, that include most importantly adequacy and sustainability of pensions. Based on these goals, the countries agreed on the set of monitoring indicators, including also the replacement ratios as well as individual replacement rates. So, the OMC did not introduce indicators (lines 85-86), rather there is an agreement on indicators that are used to monitor the goals of the OMC (one can look at the work of David 
Natali or Caroline de la Porte). 

Thank you very much for your suggestions. According to the Article 27 and Article 67 in the C102 - Social Security (Minimum Standards) Convention, 1952 (https://www.ilo.org/dyn/normlex/en/f?p=NORMLEXPUB:12100:0::NO::P12100_INSTRUMENT_ID:312247), provisions of pension benefits to typical worker shall be deemed to be satisfied as demonstrated below. It should be noted that this standard might be high for non-standard employees in developing countries.

Moreover, it is true that the OMC does not introduce specific adequacy indicators. We have corrected the statement in lines 85-86.

In text, it would be good to expand some of the conclusions related to the presented data. For example, if the expansion of the pension system to cover farmers in 2006 explains some of the drop in average pension/average wage ratio? There is definitely a very high decline in this ratio in the past few years, which seems to be caused by the high wage growth and not following this growth the pension indexation. 

This is a very important point that we have not mentioned in the paper. The growth rate of average wages from 1999 to 2018 is around 14.5% in average, which is much higher than the pension indexation rate (8%-10%) in relevant years. This could explain the declining tendency of the average replacement rate to a large extent (lines 222-225).

Presented figures show that the distribution of contributory years, but also pension benefits is not typical, it has one or more modes that are explained by the existing rules. Therefore, I would not fit the distribution function in figures 1,4 and 6 - it does not make an economic sense. 

The purpose of presenting figure 1 is to show the distribution of accumulated contribution years, namely contribution density, of different types of retirees. For the enterprise-employees, they have regular distributions; whereas for non-standard employees and land-lost farmers, they have abnormal contribution patterns which are largely caused by policy expansion regulations. The differences of contribution patterns among different retirees contribute to their final differences of pension benefits.

Figure 4 and figure 6 are the distributions of calculated individual replacement rates (at single time point of retirement) and the benefit-cost ratio (based on life-cycle) by gender accordingly. The bars are actual values and the normal distribution curves are simulated ones to show mean values.

Reviewer 2 Report

The article addresses an interesting and important topic from the social and economic point of view. In many countries, researcher's and policymaker's work are being conducted to answer the question of how pension systems should be constructed to meet the challenges of an ageing population. The problem is that the authors try to refer to the "Western" (mostly European & North American) standards of pension systems and refer to principles that have been adopted in other development conditions - countries basing their social and economic systems on democracy and market rules. The social and economic development of China is based on other principles, much more centrally controlled. This has a huge impact on the level of prices and wages, which are not regulated in a market like in the EU or US countries. In addition, the share of international pension funds is negligible, in practice there is no private pension insurance system in the European or American sense. This results in a greater dependence of pensioners on the basic (state) benefit from pension system, especially in situations where they usually do not derive income from set-aside capital (e.g. real estate purchased for pension investment purposes as it is very popular in Western countries). That is why readers may be surprised when the authors try to make analyses based on the methodology used in countries with a high-developed market economy. 
So, I think that the article is very valuable because it deals with topics that have not been discussed in this aspect by researchers in China before (as the authors themselves noted, in China researchers usually judge the reasonable level of pension benefits based on the basic living needs of the elderly after retirement). However, explanations are needed to show the differences between the Chinese pension system and Western pension systems (although it is difficult if in China policies on contribution bases are not consistent, and there are huge differences in particular provinces, as authors noted). It is also necessary for the authors to justify that the use of the same indicators as in analyses of pension systems in other countries is justified for Chinese pension system. In particular, it seems unclear what is the "basic old-age pension system" and the "residents' public pension system". The second is not discussed in this paper, what is ok, but, if it has a large role (share) in the income of retirees, then perhaps the authors should look differently at the analysed indicators of the first  (basic old-age) system. Therefore, the article should be supplemented with a few sentences explaining the importance of systems other than the "basic old-age".
There are also doubts when comparing (e.g. in Figures 2 and 3) the monthly contributions and the amount of pension benefits in yuan in such a large time interval, without taking into account inflation or the change in the purchasing power of the yuan. And here is another problem related to the fact that the yuan is not a world currency subject to market rules in the sense of the dollar or euro (of course because the yuan is not fully convertible in international market and the capital and financial market is not fully open).

The remaining empirical part of the article is good, I have only doubts about choosing only one city. However, considering the difficulties in obtaining data, this approach of the authors should be considered acceptable. The authors pointed out the limitations in this regard, and this is fine.
There are minor language errors and awkwardness in the article, including numbering of the 46 position literature that does not exist

Author Response

The article addresses an interesting and important topic from the social and economic point of view. In many countries, researcher's and policymaker's work are being conducted to answer the question of how pension systems should be constructed to meet the challenges of an ageing population. The problem is that the authors try to refer to the "Western" (mostly European & North American) standards of pension systems and refer to principles that have been adopted in other development conditions - countries basing their social and economic systems on democracy and market rules.  The social and economic development of China is based on other principles, much more centrally controlled. This has a huge impact on the level of prices and wages, which are not regulated in a market like in the EU or US countries.

In addition, the share of international pension funds is negligible, in practice there is no private pension insurance system in the European or American sense. This results in a greater dependence of pensioners on the basic (state) benefit from pension system, especially in situations where they usually do not derive income from set-aside capital (e.g. real estate purchased for pension investment purposes as it is very popular in Western countries). That is why readers may be surprised when the authors try to make analyses based on the methodology used in countries with a high-developed market economy. 

So, I think that the article is very valuable because it deals with topics that have not been discussed in this aspect by researchers in China before (as the authors themselves noted, in China researchers usually judge the reasonable level of pension benefits based on the basic living needs of the elderly after retirement). However, explanations are needed to show the differences between the Chinese pension system and Western pension systems (although it is difficult if in China policies on contribution bases are not consistent, and there are huge differences in particular provinces, as authors noted). It is also necessary for the authors to justify that the use of the same indicators as in analyses of pension systems in other countries is justified for Chinese pension system. 

Thank you very much for pointing out a very important question that whether indicators are applicable when they are used and developed in China’s context. It is true that in the past, the traditional Chinese retirement system for urban employees is based on planned economy, however, with the transition of China’s economy from planned economy to market economy in the 1980s, old-age social insurance has been established since 1998. The newly established public pension system is a Mismark Pay-as-you-go model, which belongs to the mandatory earnings-related pension system in the context of international comparison. In the series reports of “Pensions at a glance”, pension replacement rates have been calculated and compared among OECD and G20 (including China) countries. That is to say pension replacement rates indicators (especially for individual replacement rates) are comparable within China and other developed countries regardless of the political regimes. As mentioned before, the prices and wages are no longer centrally controlled, instead, they are determined by the market after reforms since 1980s. Therefore, the replacement rates of pension benefits and wages are reasonable. However, adjustments of certain indicators which are related to retirement age, relative poverty lines and average propensity to consume in China’s context are essential in the future analysis.

In addition, the occupational and private pension system is quite underdeveloped in China too. Only a tiny percentage of employees have private pension schemes.

In particular, it seems unclear what is the "basic old-age pension system" and the "residents' public pension system". The second is not discussed in this paper, what is ok, but, if it has a large role (share) in the income of retirees, then perhaps the authors should look differently at the analysed indicators of the first  (basic old-age) system. Therefore, the article should be supplemented with a few sentences explaining the importance of systems other than the "basic old-age".

Many thanks for your suggestions and we have added some sentences explaining the importance of the system (lines 144-147). The “basic old-age social insurance/ pension” refers to the mandatory earnings-related or employment-related pension system which has a history of more than 20 years; whereas residents’ public pension system is subsidized heavily by governments, and the pension benefit constitutes a very small share of pensioners’ income and therefore it is not discussed in this paper.

There are also doubts when comparing (e.g. in Figures 2 and 3) the monthly contributions and the amount of pension benefits in yuan in such a large time interval, without taking into account inflation or the change in the purchasing power of the yuan. And here is another problem related to the fact that the yuan is not a world currency subject to market rules in the sense of the dollar or euro (of course because the yuan is not fully convertible in international market and the capital and financial market is not fully open).

Figure 2 and figure 3 display descriptive statistics of the dataset. The monthly contributions and pension benefits are nominal values in corresponding years. The absolute values are not important and what is important is the ratio of them in analyzing replacement rates, in which the inflation or purchasing power factors can be eliminated. It might be a problem for the world currency issue but unfortunately we cannot solve it in this paper.

The remaining empirical part of the article is good, I have only doubts about choosing only one city. However, considering the difficulties in obtaining data, this approach of the authors should be considered acceptable. The authors pointed out the limitations in this regard, and this is fine.
There are minor language errors and awkwardness in the article, including numbering of the 46 position literature that does not exist.

Thank you for your understanding that the administrative data in City X is only available because there is a collaborative project between the local government and our academic institute.

We have corrected the mistake in 46 position of the literature which should not be existed.

Reviewer 3 Report

This is a good paper. Future work might be in the spirit of SHARE; https://academic.oup.com/ije/article/42/4/992/657275

Author Response

Thank you very much for your comment.